# Conception of a High-Level Perception and Localization System for Autonomous Driving

**DOI:** 10.3390/s22249661

**Published:** 2022-12-09

**Authors:** Xavier Dauptain, Aboubakar Koné, Damien Grolleau, Veronique Cerezo, Manuela Gennesseaux, Minh-Tan Do

**Affiliations:** 1AME-EASE, Université Gustave Eiffel, IFSTTAR, F-44344 Bouguenais, France; 2Sherpa Engineering, Site Nantes, 2 Rue Alfred Kastler, F-44307 Nantes, France

**Keywords:** perception, localization, mapping, autonomous vehicles, self-driving cars, deep learning, clustering

## Abstract

This paper describes the conception of a high level, compact, scalable, and long autonomy perception and localization system for autonomous driving applications. Our benchmark is composed of a high resolution lidar (128 channels), a stereo global shutter camera, an inertial navigation system, a time server, and an embedded computer. In addition, in order to acquire data and build multi-modal datasets, this system embeds two perception algorithms (RBNN detection, DCNN detection) and one localization algorithm (lidar-based localization) to provide real-time advanced information such as object detection and localization in challenging environments (lack of GPS). In order to train and evaluate the perception algorithms, a dataset is built from 10,000 annotated lidar frames from various drives carried out under different weather conditions and different traffic and population densities. The performances of the three algorithms are competitive with the state-of-the-art. Moreover, the processing time of these algorithms are compatible with real-time autonomous driving applications. By providing directly accurate advanced outputs, this system might significantly facilitate the work of researchers and engineers with respect to planning and control modules. Thus, this study intends to contribute to democratizing access to autonomous vehicle research platforms.

## 1. Introduction

Over the last two decades, academic research and the automotive industry have shown a strong interest in the development of advanced driver assistance systems (ADAS) and automated driving systems (ADS). ADAS and ADS are a set of technologies using multiple sensors and information processing algorithms to inform the driver or to operate a vehicle with limited or no human interactions [1,2,3]. As shown in Figure 1, the main approach for the development of such complex systems is to divide the driving task into three subtasks: (1) a perception and localization task, (2) a planning task, and (3) a control task.

The perception task aims to provide a contextual understanding of the driving scene by using exteroceptive sensors such as RGB and thermal cameras, lidar, radar, or ultrasonic sensors. The localization task refers to the ability to permanently determine the vehicle position and movement using proprioceptive sensors such as GNSS, IMU, INS, odometers, and exteroceptive sensors (cameras and lidar). The planning task consists of making decisions about the destination of the vehicle by generating trajectories allowing it to avoid static and dynamic obstacles in the road scene. The motion control task allows the vehicle to follow the planning module’s trajectory by activating the appropriate actuator [4,5,6,7].

By dividing the driving task into sub-tasks, the modular approach has the advantage of bringing together different disciplines (robotics, computer vision, vehicle dynamics, etc.). This allows it to benefit from the accumulated knowledge and major advances of these different fields. However, designing the interconnection of these modules by selecting the best inputs and outputs of each module is a complex task [8]. Moreover, the modular approach has the disadvantage of being subject to error propagation [9]. Thus, an error in the perception module can lead to an erroneous trajectory planning, which in turn can lead to a risky situation. Hence, there is a need to develop, test, and validate the algorithms of each module under controlled conditions very close to reality.

However, most perception and localization algorithms for self-driving applications are tested and evaluated on public datasets. Similarly, most path planning and control algorithms are evaluated on data from simulations [10,11]. This is due to the fact that most research teams do not have autonomous vehicle research platforms. Indeed, the development and maintenance of these platforms are complex tasks [12,13].

One answer to this problem can be found in the initiative of research teams to convert commercial vehicles into drive-by-wire vehicles. For example, the research team at the LS2N laboratory in France have converted a Renault ZOE ZE into a drive-by-wire vehicle using a kit developed in-house. This kit allows the speed, steering, brake and gear shift of the car to be controlled directly by using the vehicle CAN bus [14]. By equipping these vehicles with high-level perception and localization systems, it is possible to obtain a ready-to-use autonomous driving platform, allowing for testing and validation of algorithms under controlled conditions very close to reality.

In this work, we describe the conception of a high-level, compact, scalable, and long autonomy perception and localization system for data acquisition and algorithm testing and validation in an autonomous driving context. In addition, in order to acquire data for use in building multi-modal datasets, this system embeds two perceptions (RBNN detection, DCNN detection) algorithms and one localization (lidar based localization) algorithm to provide real time advanced information such as object detection and localization in challenging environments (i.e., lack of GPS).

As mentioned before, our object detection application is carried out using clustering and deep learning methods. As deep learning methods require a massive amount of data for training and testing purposes, a part of this work consists of building an annotated dataset. To the best of our knowledge, the literature reports very few multi-modal datasets built with a 128-channel lidar [15,16,17,18,19], of which only one is annotated [20]. This study intends to contribute to the literature in order to fill this need.

## 2. Related Works

Perception and/or localization systems can be divided in two parts: (1) low-level systems based on raw sensor outputs, and (2) high-level systems, which process raw sensor outputs using algorithms in order to provide advanced outputs [21,22,23]. It is worth noting that the existing high-level perception and/or localization systems are mainly built for UAVs (unmanned aerial vehicles).

In the autonomous driving context, the literature reports several low-level perception and localization systems, which have mainly been developed to build datasets. These systems vary in terms of their application focus, sensor setup, data format, size, and other aspects. For example, the KITTI dataset consists of data collected by four cameras, one lidar, and one GPS/IMU system [24]. Similar to the KITTI dataset, the AppolloScape, HD3, and A*3D datasets contain forward-facing cameras, lidars, and GPS/IMU systems [25,26,27]. the KAIST dataset sensor setup is similar, and include a thermal camera for night driving conditions [28], while the nuScenes, Argoverse, and Waymo datasets provide 360 FOV coverage with their camera configurations in addition to others sensors (i.e., lidar, IMU/GPS systems, and radars for the nuScenes dataset) [29,30,31]. The Oxford RoboCar Dataset has been released for localization and mapping purposes only. It includes data from one stereo-camera, three cameras, three lidars, and one GPS/IMU system; however, it contains no object annotations [32]. More recently, the CADC dataset has been designed to provide data on varying degrees of snowfall [33]. Our approach here is to develop a high level, compact, scalable, and long autonomy perception and localization system. In addition, in order to acquire data from multiple sensors to build the dataset, this system embeds three algorithms to provide real-time advanced information such as object detection or localization in challenging environments (lack of GPS) using data from exteroceptive sensors.

It is worth noting that object detection mainly involves on-road pedestrians and vehicles, traffic lights, and traffic signs. In the literature, most algorithms target the detection of objects using 2D image from cameras. These algorithms are split into conventional detection algorithms and deep learning-based methods. Most traditional detection methods are composed of two steps. First, important features are extracted from raw camera images. These features allow the valuable information to be efficiently represented, and are robust against illumination conditions, scaling of objects, and view angle. Next, after the features are made available, a learning algorithm is applied to recognize the objects within the image. More recently, deep learning methods in which feature extraction is fully and automatically integrated into the learning process from a training dataset have shown superior performance compared to conventional methods [34,35,36].

Image-based 2D object detection algorithms using deep learning methods can be classified into two categories: (1) two-stage (or region-based) detectors and (2) single-stage (or unified) detectors [37]. Two-stage algorithms such as R-CNN (region-based convolutional neural network) [38], Faster R-CNN [39], Mask R-CNN [40], and others generate a region of interest in the first stage using a region proposal network (RPN). These proposal regions are thereafter used as inputs to a second network, which regresses the exact bounding box and performs classifications. On the other hand, one-stage algorithms such as YOLO [41], SSD [42], and others treat object detection as a regression problem by extracting global features in order to predict object positions (bounding boxes) and class probabilities. In general, two-stage detectors are more accurate than one-stage detectors. However one-stage detector are faster than two-stage detectors [43]. Thus, depending on the targeted application, a trade-off has to be made between accuracy and efficiency.

Three-dimensional object detection based on point cloud representations can be split into three categories: (1) projection-based methods, (2) voxel-based methods, and (3) point-based methods [44]. In projection-based methods such as YOLO3D [45], PIXOR [46], and others, a 3D lidar or 3D camera point cloud is projected in a cylindrical way (front view projection) or in bird’s eye view to form a 2D compact map. This allows the 2D detectors cited earlier to be adapted for 3D detection tasks. Contrary to projection-based methods, voxel-based methods such as SECOND [47], VoxelNet [48], and others attempt to exploit the 3D structure of the point cloud by discretizing the 3D space into a set of voxel grids in order to predict object positions and class probabilities. Point-based methods such as PointNet [49], PointNet++ [50], and others use the native point clouds (i.e., no voxelization or projection) to directly learn features and identify objects in the point cloud data.

The next section briefly describes the sensors embedded on the platform and the software architecture that allows the whole system to work properly in real time.

## 3. Platform Description

The perception and localization system installed on an experiment vehicle is shown in Figure 2. It includes:A SBG systems inertial navigation system integrating a dual-antenna GNSS receiver for vehicle localization, inertial (acceleration, angular velocity), and attitude (roll, pitch, yaw) measurements.Two FLIR global shutter cameras coupled with 90° Fujinon lenses to capture images of the front half of the vehicle; each camera-lens pair is installed in an autoVimation IP67 waterproof housing for protection against splashing and dust.An Ouster 128-channel rotating lidar with a vertical field of view and vertical resolution of 45° and 0.35°, providing a 3D point cloud mapping of the vehicle’s environment.A GPS time server for sensor data timestamping.Two ethernet connections for data transfer.A ruggedized embedded computer based on Nvidia Jetson AGX, with an expanded memory using a 4 TB SSD device for data storage.

Except for the power supply battery inside the vehicle, all platform components are installed on a mechanically screwed structure coupled with an ASA (Acrylonitrile Styrene Acrylate) shell. The latter protects the components of the system from all solid projectiles (stones, hail, insects) while ensuring good heat dissipation. The entire system is powered by an external lithium iron phosphate battery (LiFePO4) with an integrated BMS (Battery Management System). This allows the system to be energy self-sufficient and to operate safely. In standard operation (Table 1), the system’s autonomy easily covers a 12-h day.

Figure 3 shows how sensor data and time synchronization are distributed across the platform. Two types of information are defined: (1) data from the sensors and (2) information related to the synchronization and data timestamping. For data acquisition, transfer, compression, saving, and visualization, we use an ROS open-source framework. ROS is a flexible development middleware available as an overlay on Debian-based Linux distribution. It provides various tools and libraries to facilitate the design of software, mainly for robotics systems.

The perception systems are mostly composed of lidars and vision systems (RGB camera and/or infrared camera), providing complementary information of the observed scene. Cameras provide details on the texture of objects, while lidars provide accurate depth information. Merging these two types of information can significantly improve the performance of classification algorithms or build very accurate maps. The efficient use of information from different sources requires precise synchronization and calibration of the sensors involved. The followings segments describe the platform sensor synchronization and calibration proceedings.

### 3.1. Sensor Synchronization

In order to guarantee accurate interpretation of the driving scene, all sensor acquisition is carried out in a synchronized manner and dated on a common time base.

As shown in Figure 4, a pulse signal generated by the lidar is used to trigger the acquisition of the cameras through the electronic card, when the lidar beam passes between the two cameras. In addition, the lidar has a PTP (Precision Time Protocol) synchronization input. The latter is connected to the output of the time server, and allows the cameras and lidar data to be time-stamped using the international atomic time (TAI). The time-stamps of these data are then brought back to the coordinated universal time (UTC) by subtracting the leap seconds separating the TAI from the UTC times. In addition, the inertial measurement unit provides the GNSS and IMU data directly synchronized and time-stamped in UTC. As shown in Figure 5, this configuration ensures the temporal synchronization of all sensors of the perception and localization system.

### 3.2. Sensors Calibration

Any optical system composed of a camera or a camera coupled with a lens produces images with aberrations (radial and tangential distortions) due to the properties of the lens and the manufacturing defects. These distortions can significantly alter the rendering of the captured scene by giving a curvilinear aspect to linear objects. The two types of distortion most commonly encountered are: (1) radial distortion (barrel distortion), characterized by a curvature of the straight lines outwards from the center of the image, and (2) tangential distortion (pincushion distortion), characterized by curvature of the straight lines towards the center of the image. These aberrations can be corrected using the Brown-Contrary or Kanalla–Brandt camera distortion models to determine the distortion coefficients, (k_1_, k_2_, k_3_) and (p_1_, p_2_), respectively, of the radial and the tangential distortion coefficients [51,52]. These parameters are determined by carrying out a series of acquisitions of a pattern with known dimensions (checkerboard) under different points of view.

On the other hand, considering each lidar/camera combination as a unit of independent sensors, the extrinsic calibration consists in finding the rigid transformation between the camera and lidar coordinate systems. This rigid transformation, represented by a rotation matrix (R) and a translation vector (T), defines the extrinsic parameters of a lidar–camera system, and the objective of the extrinsic calibration procedure is to estimate them. Thus, similar to intrinsic calibration, a checkerboard can be used to simultaneously detect points of interest in the camera and lidar coordinate system frame. Then, the optimal rigid transformation can be determined to match the two sets of points. Using the extrinsic calibration parameters, the intrinsic distortion coefficients, and intrinsic camera parameters such as focal length (fx, fy) and optical centers (cx, cy), it become possible to project the lidar cloud point cloud onto camera images (Figure 6).

## 4. Dataset

We used the perception and location system to collect data in order to build a dataset. The drives were carried out in the center and outskirts of Nantes between March and October 2021. In order to obtain a varied data set, these drives were carried out under different weather conditions and different traffic and population densities. In compliance with data protection regulations (RGPD), the acquired data were anonymized. One hundred and two (102) continuous sequences of 10 s each were selected from the different acquisitions and then annotated by an annotation partner. Within each sequence, about 100 synchronous lidar/camera/GNSS/IMU samples formed a temporally coherent sequence. It is worth noting that the images were rectified and the lidar point clouds corrected for deformation due to the ego-vehicle movement. The class distribution is shown in Figure 7. This dataset is composed of about 10,000 annotated lidar samples, which corresponds to about 130,000 boxes. The dataset was used to train and evaluate the performance of the perception algorithms. The results of these studies are presented in the next section.

## 5. Embedded Algorithms

In order to achieve high-level perception and localization tasks, three algorithms were developed and embedded within the system: (1) a lidar data-based clustering analysis, (2) a lidar-based deep learning classification and detection algorithm, and (3) a lidar based localization algorithm. In the following, we briefly describe the functioning of these three algorithms.

### 5.1. Clustering Analysis by RBNN (Radially-Bounded Nearest Neighbor)

Clustering analysis aims to divide sample data into a set of distinct groups in an unsupervised way, that is, without learning. Each group (cluster) is built from points sufficiently close to each other with respect to the points of the other clusters. The clustering algorithm embedded in the perception and localization system is based on a radially-bounded nearest neighbor (RBNN) strategy [53,54,55]. As shown in Figure 8, the algorithm takes the raw lidar data as input and provides the convex hull of the objects within the road scene as outputs.

Unlike other clustering algorithms, such as K-Means, agglomerative clustering, and Gaussian mixture model, RBNN does not require a priori knowledge of the number of objects contained in the scene. Indeed, the main hyperparameters to tune are: (1) the minimal number of points for a cluster to be considered as valid and (2) the radius search. We implemented the RBNN algorithm using a Kd-tree search structure. The processing time is less than 20 ms with a 10 cm sub-sampling grid, which is consistent with real time self-driving applications.

The performance of the clustering algorithm was evaluated using the dataset (described in Section 3) by determining the true positive rate (TPR). An object was considered as correctly detected if the cluster satisfied two requirements: (1) the ratio of the intersection of the convex hull and the ground truth (from the dataset) on the convex hull was greater than the threshold of 0.1, and (2) the cluster had at least 20 lidar impact points. Otherwise, the object was considered a false negative.

Table 2 shows the detection results by distance intervals. It can be observed that the detection rate is maximal when the objects are close to the lidar. Indeed, a detection rate of 98.76% is obtained for objects located within a radius of 10 m around the lidar. However, this value decreases as objects become farther from the lidar. Indeed, the detection rate drops to 80% for objects located between 40 and 50 m from the lidar.

### 5.2. Object Detection by DCNN (Deep Convolutional Neural Network)

Unlike clustering algorithms, deep learning-based detection methods aim to predict the content of a scene in a supervised way (with learning). The detection algorithm embedded in the perception and localization system is based on a deep convolutional neural network architecture. As shown in Figure 9, the algorithm takes lidar data as input and provides three-dimensional bounding boxes of the objects within the road scene as outputs. Each predicted box is described by nine parameters (u, v, d, w, l, h, α, c, s):(u, v, d): represent the center of the box(w, l, h): represent the 3D dimensions of the boxα: the orientation of the box in top view (yaw angle)c: the class of the box (type of object)s: the confidence score associated with the prediction

For this study, our preference was for a voxel-based method over a bird’s eye view (BEV) projection, as we wanted to preserve the initial 3D structure of the lidar point cloud. The input of the DCNN algorithm consists of fixed-size voxel grid combined with a 1 × 1 convolution window. This results in representation of the lidar data in pillar form, which is well known to improve backbone efficiency. It follows a PointPillars backbone and a single shot detection (SSD) head detection with output encoding similar to CenterPoint [56,57]. This single-stage architecture (Figure 10) allows for a good trade-off between run-time performance and prediction quality.

The DCNN was trained to classify objects of three types: (1) passenger vehicles (car), (2) medium vehicles (van, pickup, minivan), and (3) large vehicles (bus, truck). The dataset containing 10,000 lidar scans was split into a training set of 7500 samples and a validation set of 2500 samples. The network was trained for 200 epochs by a single GPU with two samples per batch, and the Adam optimizer was adopted. The learning rate was $1\times 10^{-4}$ and the loss functions were focal loss for the classification task (c, s) and smooth L1 for the regression task (u, v, d, w, l, h, α). Using Nvidia’s Tensor Rt framework and mixed precision (FP16) inference optimization, the network achieved a total prediction time of less than 30 ms with an inference time of about 10–15 ms. 

The DCNN detection performance was evaluated by determining the true positive rate (TPR, Equation (1)) and the false discovery rate (FDR, Equation (2)) for both distance intervals. TPR and FDR are estimated using the intersection on union (IoU) of the prediction box over the ground truth annotation:(1)$$TPR=\frac{TP}{TP+FN}$$
(2)$$FDR=\frac{FP}{FP+TP}$$

Here, TP, FP, and FN are, respectively, true positive predictions, false positive predictions, and false negative predictions.

The results (Figure 11) show good detection between 2–20 m (around 95% for TPR and 6% for FDR) for cars and medium vehicles, then the detection performance decreases beyond 20 m. The same trend can be seen for large vehicles (truck, bus, etc.) though with lower detection performance. This is due to the fact that the last class includes objects with varied sizes and which are under-represented in the dataset. This makes the learning process more difficult for this class.

A detailed analysis of the positioning, size, and orientation errors of the predictions confirms the impact of distance on the detection quality. As shown on Figure 12, while the estimates are fairly accurate, the standard deviation for each prediction parameter increases with distance. This is particularly the case for orientation, for which the estimation may be valid modulo 90 deg due to an ambiguity that the network is unable to resolve. A solution may be to inject the lidar intensity signal as input for the DCNN. This could help the network to find the correct orientation by using the reflective properties of the vehicle’s license plates.

Table 3 presents the average precision (AP) calculated individually for each class using the area under the precision–recall curve (AUC-PR) for an IoU threshold of 0.70. Table 4 shows that the medium average precision (mAP) for cars and medium vehicles is similar to the values reported in the literature. 

### 5.3. Lidar-Based Loccalization

Localization refers to a mechanism for determining the position of an object on a map. Depending on the technology used, the positioning accuracy can vary from meters to a few centimeters. In a navigation context, the most commonly used technology is based on a satellite system called GNSS (global navigation system satellites). However, positioning by satellite has several limitations. Meteorological conditions can affect satellite visibility, resulting in weak positional accuracy. In addition, satellite signals may not be available inside structures such as in a building, garage, parking, structure, or tunnel. This makes satellite positioning systems alone incompatible with autonomous driving applications.

Exteroceptive localization algorithms aim to provide a continuous and accurate position service using a pre-built map. The map building process consists of creating probabilistic occupancy grid submaps (Figure 13a) using lidar, inertial, and GNSS data. Each submap is a discretization of the environment into a set of two-dimensional regular cells characterized by assigning a probability value that a cell will be occupied by an obstacle. More specifically, local submaps are built by iteratively adding lidar scans in a fixed reference frame. The successive poses of the lidar are obtained using a Monte Carlo localization filter, and the large drift inherent to long-term exploration is avoided using GNSS injection and loop closures inside a pose graph optimization. After the map has been built, essentially using lidar and inertial measurement data, localization is performed by correlating an actual lidar scan to the map (Figure 13b). The algorithm operation can be summarized as follows:

Step 1 (Initialization)—the algorithm randomly generates a number N (N = 500 in our application) of discrete poses (Equation (3)). Each pose (called a particle) is a possible location of the vehicle within the map. Because there is an equal probability that the vehicle is anywhere on the map and facing any direction, the poses have the same weight ($w_{i}$).
(3)$$q_{i}=\left[\begin{array}{c} x_{i}\\ y_{i}\\ \theta_{i} \end{array}\right],\;i=1,\;\ldots ,\;N$$Here, ($x_{i},\;y_{i}$) and $\theta_{i}$ are the 2D coordinates and the orientation of the “*i*” pose, respectively.Step 2—Using the speed (v) and the yaw rate (φ) of the vehicle (from the IMU sensor), the algorithm moves all the particles using the motion model in Equation (4).
(4)$$q_{i}\left(t+dt\right)=q_{i}\left(t\right)+\left[\begin{array}{c} \frac{v}{\varphi }\left(-\sin\theta_{i}+\sin\left(\theta_{i}+\varphi *dt\right)\right)\\ \frac{v}{\varphi }\left(\cos\theta_{i}-\cos\left(\theta_{i}+\varphi *dt\right)\right)\\ \theta_{i}+\varphi *dt \end{array}\right]$$
where dt is the time between two IMU measurements.Step 3—When a lidar scan becomes available, for each pose (particle) the algorithm calculates the relative likelihood between the actual lidar scan and the point cloud data of the map surrounding the pose. Then, it updates the weight of each pose.Step 4—The particles are resampled using a low-variance resampling algorithm.Step 5—The particles are used to estimate the vehicle’s pose by determining the centroid of the particles (Equation (5)) after weight normalization (Equation (6)).
(5)$$\left[\begin{array}{c} x\\ y\\ \theta \end{array}\right]=\left[\begin{array}{c} \overset{N}{\underset{j=1}{\sum }}w*\;x_{j}\\ \overset{N}{\underset{j=1}{\sum }}w*\;y_{j}\\ atan2\left(\overset{N}{\underset{j=1}{\sum }}w*\;\sin\theta_{j}\;,\;\;\overset{N}{\underset{j=1}{\sum }}w*\cos\theta_{j}\right) \end{array}\right]$$
(6)$$w=\frac{1}{N}*\sum_{j=1}^{N}w_{j}$$Step 6—Return to Step 2.

The processing time of the embedded localization algorithm is about 15 ms. As shown in Table 5, using the lidar localization algorithm we are able to attain an accuracy of a few centimeters.

## 6. Discussion

Building a high-level perception and localization system is a complex task that includes both hardware and software components. The hardware part is not limited to putting together the sensors; indeed, it implies that requirements such as calibration, synchronization, and timestamping must be satisfied. In this regard, our system is similar to the existing ones in the literature. However, we focused on system autonomy in term of energy consumption and memory stockage, on the one hand, and system usability and extensibility on the other. Thus, our system can be installed or uninstalled on different vehicles without breaking the exteroceptive calibration, and new sensors can be added while maintaining the same software architecture.

The other part of this work consist in building perception and localization algorithms in order to provide advanced outputs. Using our constructed dataset, the performances of the perception algorithms were evaluated considering the distance factor. The RBNN method has the advantage of not requiring training data, and the algorithm reaches more than 97% correct detection in the 20 first meters around the vehicle. Nevertheless, clustering does not provide any information on the type of target detected. On the other hand, the DCNN algorithm shows good detection for the same distances (around 95% for TPR and 6% for FDR and 0.87 for AP) for cars and medium vehicles and lower detection performance for large vehicles (truck, bus, etc.). DCNN detection performance could be improved with a larger dataset or by fusing lidar and camera data for multi-modal object detection [58,59,60,61]. Nevertheless, the perception algorithms do not provide a full description of the vehicle environment. Indeed, the DCNN was not trained to detect pedestrians and cyclists. In addition, the perception task does not provide traffic sign and drivable surface detection. The next step for the perception task is to develop and validate these algorithms.

Lastly, the localization algorithm is able to reach centimetric accuracy using a pre-build card. However, 2D localization algorithms have limitations, particularly with respect to certain types of building such as multi-level parking structures. Upgrading this 2D algorithm to a 3D solution would lead to a more complete and robust localization tool.

## 7. Conclusions

This paper presents the concept of a high-level perception and localization system composed of a 128-channel lidar, two global shutter cameras, an INS (IMU + GNSS), a time server, and an embedded computer. In addition, the system embeds three algorithms which process raw sensors outputs and provide real time advanced outputs, such as objects in road scene or localization in challenging environments. The performance of the three algorithms are competitive with the state-of-the-art. Moreover, the processing times of these algorithms are compatible with real-time autonomous driving applications. However, the perception task needs to be enriched by traffic signs and drivable surface detection algorithms. Similarly, upgrading the 2D exteroceptive localization algorithm into a 3D solution can lead to a more complete and robust localization tool.

After the system reaches maturity, it might be possible to equip drive-by-wire vehicles, thereby converting the latter into autonomous vehicle research platforms.

The literature reports very few annotated multi-modal datasets built with a 128-channel lidar. The present study contributes to filling this need.

## Figures and Tables

**Figure 1 sensors-22-09661-f001:**
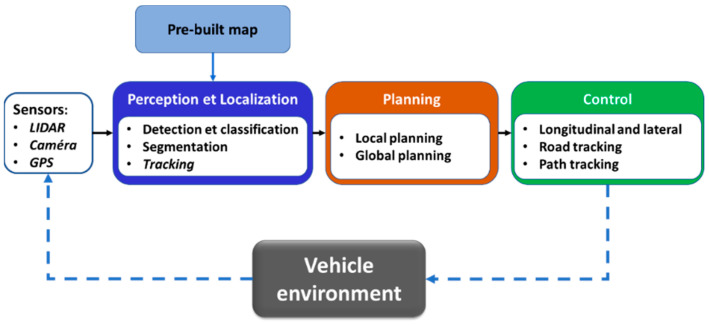
Modular architecture of automated driving system.

**Figure 2 sensors-22-09661-f002:**
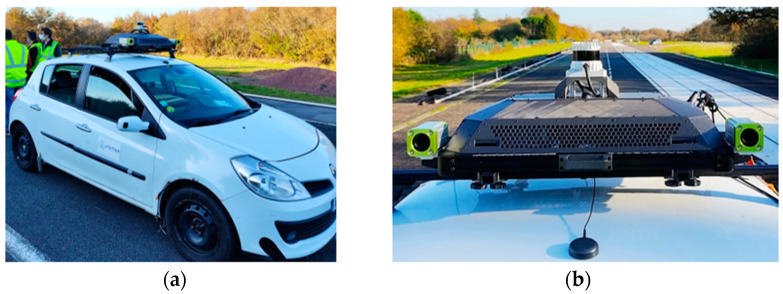
(**a**) Perception and localization system installed on an experimental vehicle; (**b**) zoom of the PLS.

**Figure 3 sensors-22-09661-f003:**
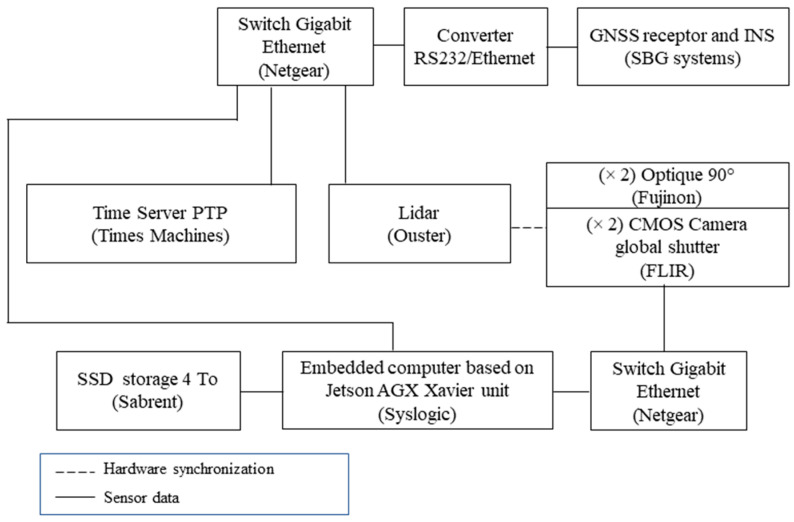
Data and synchronization communication scheme.

**Figure 4 sensors-22-09661-f004:**
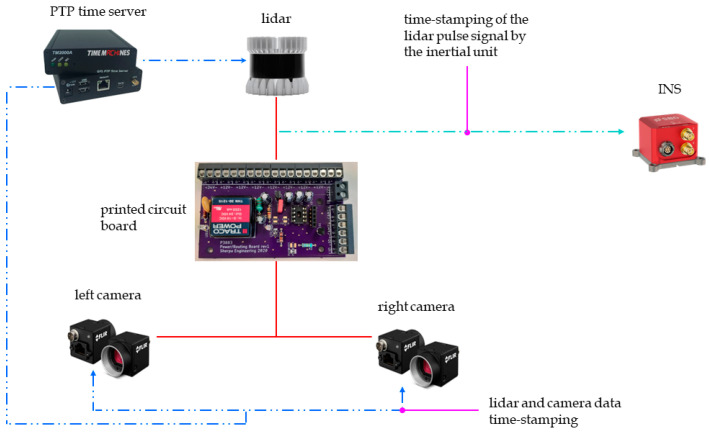
Data synchronization and time-stamping scheme.

**Figure 5 sensors-22-09661-f005:**
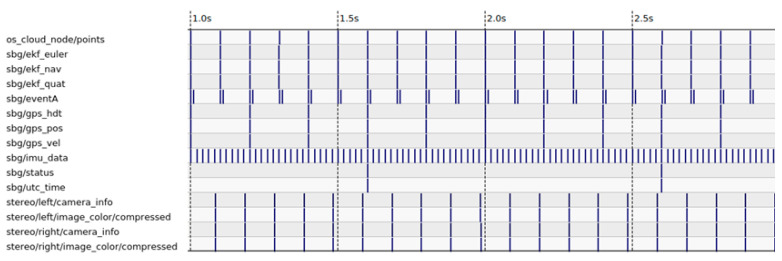
Timeline of data acquisition from the perception and localization platform’s sensors.

**Figure 6 sensors-22-09661-f006:**
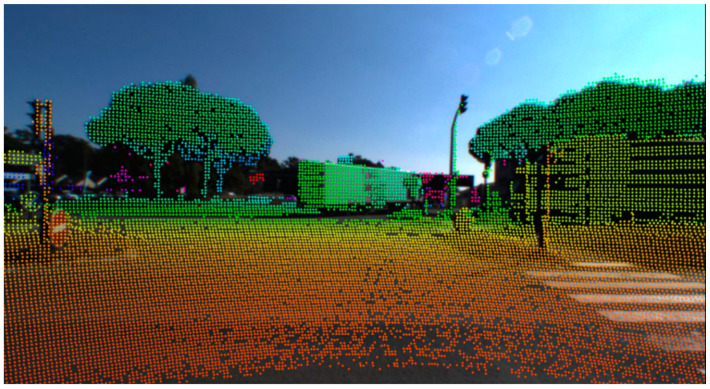
Lidar point cloud projected onto camera images.

**Figure 7 sensors-22-09661-f007:**
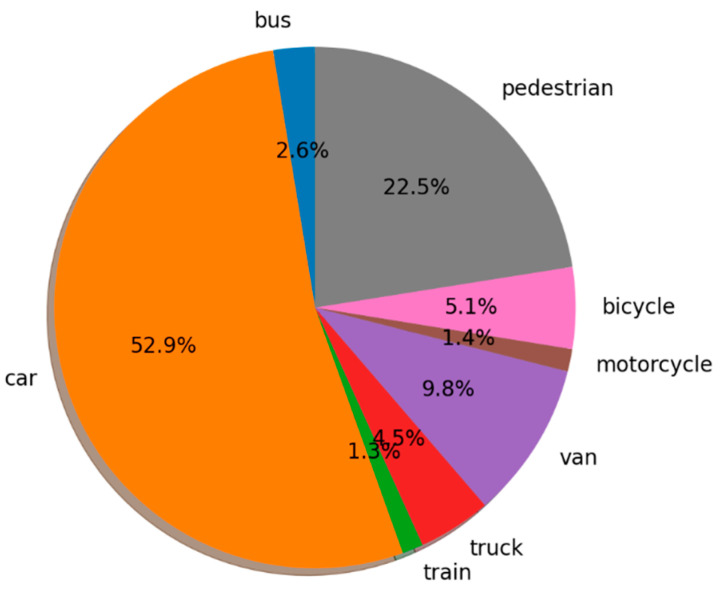
Class distribution.

**Figure 8 sensors-22-09661-f008:**
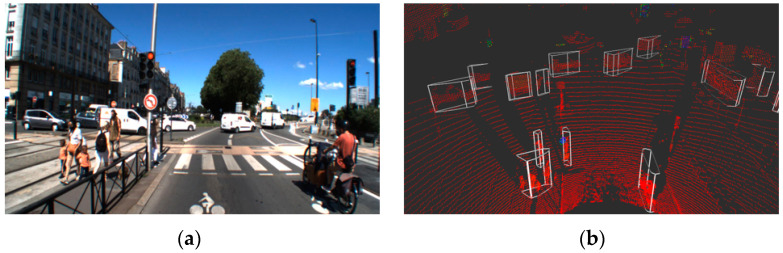
Clustering analysis: (**a**) camera image for illustration and (**b**) results of lidar clustering detection.

**Figure 9 sensors-22-09661-f009:**
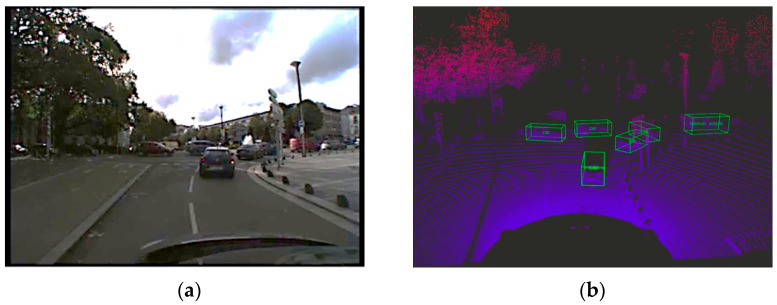
Object detection by DCNN: (**a**) camera image for illustration and (**b**) result of lidar DCNN detection.

**Figure 10 sensors-22-09661-f010:**
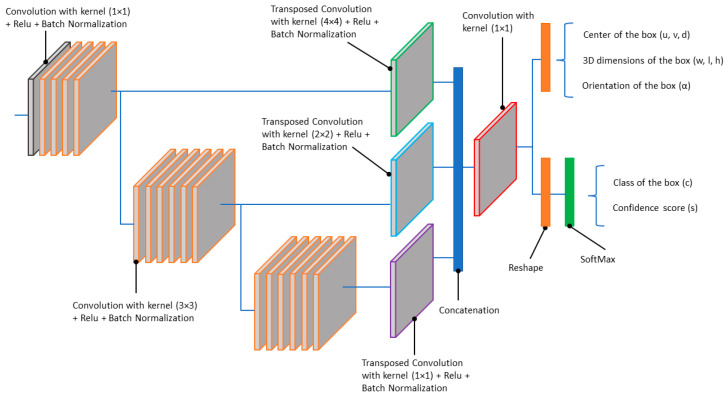
The DCNN architecture.

**Figure 11 sensors-22-09661-f011:**
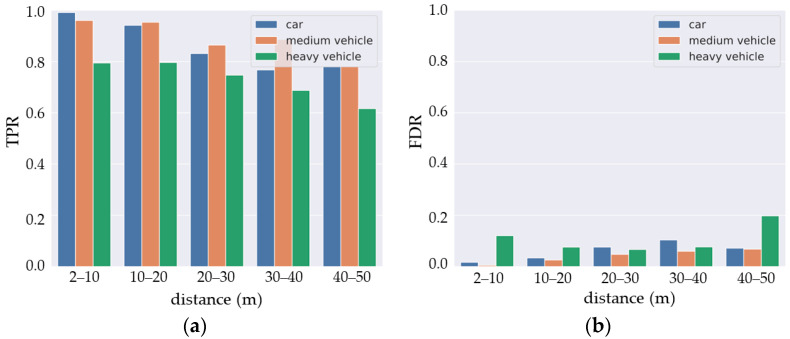
DCNN detection results: (**a**) true positive rate and (**b**) false discovery rate.

**Figure 12 sensors-22-09661-f012:**
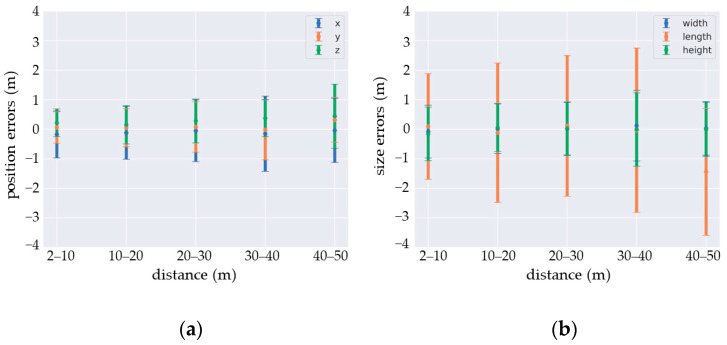
Estimation errors: (**a**) position and (**b**) size.

**Figure 13 sensors-22-09661-f013:**
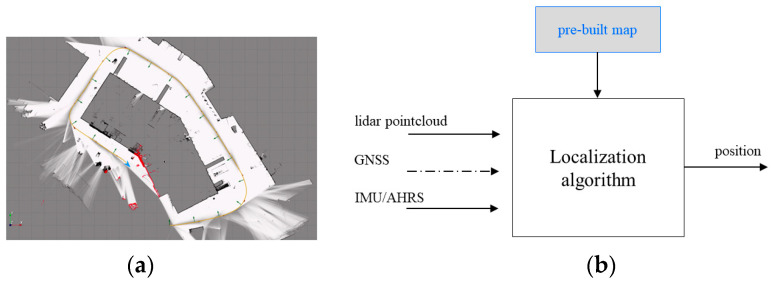
Lidar-based localization algorithm: (**a**) map building process and (**b**) localization operating schemes.

**Table 1 sensors-22-09661-t001:** Sensor frequencies in standard operating mode.

Sensors	Frequencies (Hz)
Lidar	10
GPS (INS)	10
IMU (INS)	50
Time Server PTP	1
Camera	10

**Table 2 sensors-22-09661-t002:** Clustering detection results.

Distances (m)	2 to 10	10 to 20	20 to 30	30 to 40	40 to 50
TPR (%)	98.76	97.69	94.79	89.38	80.85

**Table 3 sensors-22-09661-t003:** Average precision results.

Vehicles	Car	Medium	Big Vehicle
AP	0.87	0.87	0.71

**Table 4 sensors-22-09661-t004:** Medium average precision results.

Methods	mAP (Moderate)
Our	0.87
PointPillars [57]	0.86
PIXOR++ [46]	0.83
VoxelNet [48]	0.84
SECOND [47]	0.79

**Table 5 sensors-22-09661-t005:** Localization algorithm performance. The performance of the algorithm was evaluated by comparing the path from localization to the RTK position.

Errors	Lateral Mean Error	Lateral Mean Std	Longitudinal Mean Error	Longitudinal Mean Std
Values (cm)	1.6	5.3	5.0	9.5

## Data Availability

The data presented in this study are available on request from the corresponding author. The data are not publicly available due to confidentiality agreement within the framework of ENA project.

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
