# Peer review of "Conception of a High-Level Perception and Localization System for Autonomous Driving"

_sensors, 2022, doi:10.3390/s22249661_

Round 1

Reviewer 1 Report

·        The introduction should be greatly improved because it does not have enough information. For example, what problem do the authors want to solve? Are there any existing solutions? Which is the best and why? What is the main limitation of the best and existing approaches? What do you hope to change or propose to make it better?

·        The literature review of this paper is not enough.

·         The conclusion is shallow, and it needs to be improved.  

Author Response

A new version of the manuscrit has been submitted

Reviewer 2 Report

The reviewer recommends the article for major revision. Full comments are given in the attached PDF file.

Author Response

A new version of the manuscrit has been submnited

Reviewer 3 Report

This paper proposes a multi-modal dataset and embeds two perceptions and one localization algorithm to provide advanced real-time information. The performances of the proposed algorithms are competitive with state of the art.

Strengths:

1. The proposed method detects the object in real-time with clustering and deep learning methods using 128 channels of lidar data.

2. It also builds an annotated dataset.

 Weakness:

1. The abstract is not well-constructed; it should state the research topic of the paper. In addition, you should also note what is original in your article.

2. One of the investigation's main objectives is high-level perception; therefore, in the Introduction section, you should state the nature and scope of the research aiming from the perspective of high-level perception.

3. The introduction section does not describe the investigation's motivation and the problem clearly.

4. In the Embedded algorithms section, how do the results compare with earlier work? What is new and significant? It would be best if you described this in this section.

5. To describe the method, for example, DCNN based object detection method, you should propose details that are clear enough for competent readers to reproduce the experiments.

6. In the Discussion section, I suggest you state the unanswered questions and outline future research.

Author Response

A new version of the manuscrit has been submittted 

Round 2

Reviewer 1 Report

I've noticed an improvement in this version of the paper. However, the "Related Works" section is short and not sufficient . Please review more papers related to the topic. 

Author Response

  1. In this version of the article, we describe 2D and 3D object detection based on deep learing.
  2. References on the most important algorithms were added in the text 

Reviewer 2 Report

The authors addressed all reviewer's comments. The article is recommended for publication.

Author Response

Dear Reviewer, thank you for your positive feedback. We have been asked to improve the "related works" section.

  1. In this version of the article, we describe 2D and 3D object detection based on deep learing.
  2. References on the most important algorithms were added in the text 

Reviewer 3 Report

The authors have addressed my concerns.

Author Response

(The authors gave the same response as above.)
